# Outcome of Elective Checkpoint Inhibitor Discontinuation in Patients with Metastatic Melanoma Who Achieved a Complete Remission: Real-World Data

**DOI:** 10.3390/biomedicines10051144

**Published:** 2022-05-16

**Authors:** Leanne Perez, Wolfram Samlowski, Ruby Lopez-Flores

**Affiliations:** 1University of Nevada School of Medicine, Reno, NV 89557, USA; leannep@med.unr.edu (L.P.); rlopezflores@med.unr.edu (R.L.-F.); 2Comprehensive Cancer Centers of Nevada, Las Vegas, NV 89148, USA; 3Department of Internal Medicine, Las Vegas Kerkorian School of Medicine, University of Nevada, Las Vegas, NV 89102, USA

**Keywords:** ipilimumab, nivolumab, pembrolizumab, complete response

## Abstract

Checkpoint inhibitor therapy for metastatic melanoma has dramatically improved outcomes. Currently, 20 to 40% of treated patients achieve lengthy remissions. It is not clear whether patients in remission require ongoing therapy or if treatment can be safely discontinued. A retrospective chart review was performed of patients who underwent elective treatment discontinuation after two negative scans three months apart. Of 132 checkpoint inhibitor-treated patients, 46 achieved a complete response (34.8%) and electively discontinued therapy. The progression-free survival was 97.5% at 1 year and 94.7% at 3 years following treatment discontinuation. The median duration of follow-up was 26 months. In total, 4 of 46 individuals (8.7%) eventually relapsed (median time to relapse: 27 months). The median disease-specific survival of the entire cohort was not reached and was 100% at 4 years from the start of therapy. Two patients eventually died, one from melanoma and the other from an unrelated illness. We have identified an elective treatment discontinuation strategy that is generalizable to a variety of checkpoint inhibitor ± targeted therapy regimens. We found that most complete remissions remained durable after elective treatment discontinuation. We hypothesize that this approach could decrease potential drug toxicities, reduce the treatment-related financial burden, and improve patients’ quality of life.

## 1. Introduction

Melanoma is a potentially aggressive cancer derived from melanocytes that has a high risk of metastasis to lymph nodes and other organs if not identified and treated early [1]. It is estimated that 106,110 cases of invasive cutaneous melanoma were diagnosed in 2021, resulting in 7180 deaths in the United States [2].

Fortunately, recent advances in cancer therapy have dramatically changed the outlook for patients with metastatic melanoma. Only 10–15 years ago, the 5-year survival rate of individuals with metastatic melanoma was under 10%, with a median overall survival of only 5.3–7.7 months [3,4]. Only 15–25% of patients survived 1 year [4,5]. By 2019, the 5-year survival rate had increased to over 50% due to the development of immune check point inhibitors (CKIs) and targeted therapy agents. Single-agent anti-CTLA-4 therapy (i.e., ipilimumab) resulted in a median overall survival of 19.9 months and a 5-year survival rate of 26% [6]. Single-agent anti-PD-1-directed therapy using nivolumab resulted in a median survival of 36.9 months and a 5-year survival rate of 44% [6]. Pembrolizumab, another anti-PD-1 agent, resulted in a median survival of 23.8 months and a 5-year survival rate of 34% [7,8]. Combined anti-CTLA-4 plus anti-PD-1 therapy resulted in a 5-year survival rate of 52%, with a median survival that had not been reached [6].

In addition to CKIs, a better understanding of the spectrum of recurrent oncogenic driver mutations in melanoma has resulted in the emergence of effective targeted therapies (TT) [9]. Approximately 40–50% of melanomas harbor a *BRAF V600* mutation and, thus, may respond to treatment with BRAF ± MEK inhibitors [9]. Treatment of *BRAF-V600E*-expressing melanomas with combined BRAF/MEK inhibitor therapy (dabrafenib plus trametinib) has resulted in a 5-year survival rate of 34% in *BRAF* mutant metastatic melanoma [10]. Similar 5-year outcomes have recently been published using vemurafenib and cobimetinib [11]. However, progression-free survival with targeted therapy was only about 19% at 5 years in both studies, despite continual drug administration.

A substantial fraction of patients treated with CKI therapy achieve a durable remission. However, it is unclear how long patients need to be treated with CKIs prior to treatment discontinuation to maintain these remissions. Early clinical trials arbitrarily mandated 1 or 2 years of fixed-duration treatment prior to drug discontinuation, in the absence of significant toxicity [7]. In the Checkmate 067 trial, a planned maintenance nivolumab treatment was continued until disease progression, the occurrence of unacceptable toxic events, or patient choice [12]. It should also be noted that some patients who enrolled in these trials discontinued treatment due to toxicity and still achieved durable remissions.

Although there is no standardized approach to the duration of CKI treatment, it is apparent that protracted treatment increases the financial burden to patients and payers. The duration of treatment may increase the risk of autoimmune toxicity. Lengthy treatment also has a significant impact on patient quality of life. We describe our experience with systematic treatment discontinuation in a community oncology setting.

## 2. Materials and Methods

A retrospective chart review of patients with metastatic melanoma treated by a single physician (WS) at the Comprehensive Cancer Centers of Nevada between 2015 and 2021 was performed. A secure HIPAA-compliant iKnowMed database (McKesson, Houston, TX, USA) was searched for patients with metastatic melanoma who had received an initial treatment with checkpoint inhibitor therapy (pembrolizumab, nivolumab, or ipilimumab) for metastatic or unresectable malignant melanoma. Individual records were then reviewed to identify patients who had achieved a complete remission and then underwent a planned discontinuation of CKI therapy. Patients who had not achieved a complete response, who had not yet completed planned treatment, or who had treatment discontinuation for toxicity were excluded from the analysis. Patients receiving adjuvant therapy for completely resected metastatic disease were also excluded.

Data from eligible patients were extracted into a spreadsheet (Excel, Microsoft, Redmond, WA, USA) for analysis. All patient-identifying information was removed. The data acquired for each patient included: a unique (assigned) patient ID, age, gender, and melanoma primary and metastatic sites as well as comorbid medical conditions. The CKI agent employed, CKI dose, CKI start date, CKI end date, and the number of CKI doses were recorded. The date of progression, CKI toxicity, and any other treatments were extracted from the computer record. The objective response assessment, duration of response, and any acute or chronic toxicity resulting from therapy were also recorded. If applicable, the cause of death was recorded. This study design was reviewed by the Western Institutional Review Board (IRB) and was deemed to be exempt from a full IRB review.

The patient response to initial CKI therapy was measured from the start of the effective CKI regimen that induced a complete remission. The maximal objective response rate (at >9 months) was determined using the RECIST 1.1 criteria [13]. A complete response (CR) was defined as the disappearance of all target and non-target lesions and the normalization of tumor marker levels. A partial response (PR) was defined as more than a 30% reduction in the sum of bidimensional tumor measurements. Progressive disease (PD) was described as a >20% increase in the sum of bidimensional tumor measurements or the development of new metastases. Stable disease (SD) was defined as any response not meeting the criteria for CR, PR, or PD. Toxicity was graded using the CTCAE 4.0 criteria [14]. In some patients with stable residual radiographic abnormalities, core needle biopsies were performed, if technically feasible, to verify histologic complete remission if there was a prolonged partial response or stable disease.

### 2.1. Treatment Regimens

Melanoma treatment regimens have evolved over the span of this study. Thus, patients were treated with a variety of intravenous regimens: nivolumab (3 mg/kg, or a fixed dose of 240 mg every 2 weeks, or alternatively 480 mg every 4 weeks), pembrolizumab (2 mg/kg every 3 weeks or a fixed dose of 200 mg every 3 weeks), or combined therapy with ipilimumab (either 1 or 3 mg/kg) plus nivolumab (either 3 or 1 mg/kg) every 3 weeks for four doses, with subsequent nivolumab maintenance [15]. A small number of patients who progressed on CKI therapy and had a targetable mutation (*BRAF*, *NRAS*, or *NF1*) were offered the cautious addition of a low-dose targeted therapy (TT) with the continuation of PD-1 antibody therapy. TT typically consisted of dabrafenib 75 mg/day, encorafenib 75 mg/day, trametinib 1 mg/day, or binimetinib 15 mg b.i.d. If there was no apparent toxicity after a week of concurrent therapy, a cautious dose escalation of BRAF or MEK inhibitors was considered [16]. Patients were closely monitored for signs of toxicity prior to each treatment, and CKI therapy was interrupted if toxicity persisted or could not be controlled with a dose reduction or adjunctive treatment.

### 2.2. Statistical Analyses

Descriptive statistics were calculated via an Excel spreadsheet (expressed as data ranges, medians, and standard deviations). The overall survival and progression-free survival were calculated from the start of CKI therapy to the last visit if the patient was still alive or to the date of death. The progression-free survival from treatment discontinuation was also calculated. A Kaplan-Meier analysis was performed to evaluate the progression-free and overall survival comparing patients achieving a CR with those not achieving a CR [17]. A statistical comparison of these groups was performed by a log rank test [18]. The data analysis cutoff date was 31 July 2021.

## 3. Results

### 3.1. Patient Demographics

A total of 132 patients diagnosed with metastatic melanoma initially treated with immunotherapy were identified. Patients were treated with a variety of agents based on regulatory drug approvals at the time of their diagnosis. Forty-six individuals achieved a complete remission (34.8%) and discontinued treatment after two negative scans, as previously described (Table 1). In rare patients with persistent stable radiologic abnormalities, biopsies were performed to verify pathologic complete responses. Of the 46 patients who achieved a CR, there were 32 men (69.6%) and 14 women (30.4%). The youngest patient was 25 years old at time of diagnosis, while the oldest patient was 83 years old. The median age at the time of diagnosis was 66 (±12 years standard deviation) years. Two patients identified as Hispanic (4.3%) and the remaining forty-four patients identified as Caucasian (95.7%) (Table 1).

All patients were treatment-naïve at the onset of therapy. Forty-one patients had stage IV melanoma (89.1%) at the start of immunotherapy treatment. The remaining five individuals had unresectable T4 or stage III melanoma (10.9%). The melanoma primary sites were trunk (32.6%), leg (17.4%), unknown primary (10.9%), scalp (8.7%), face (8.7%), arm (6.5%), sinonasal (6.5%), ear (4.3%), neck (2.2%), and subungual (2.2%).

### 3.2. Treatment

Ten patients received nivolumab monotherapy (21.7%). Thirteen patients were treated with pembrolizumab monotherapy (28.3%). Twenty-three were treated with combination therapy: eighteen were treated with a standard-dose ipilimumab (3 mg/kg) plus nivolumab (1 mg/kg) regimen (39.1%), while five patients received the alternate-dose ipilimumab (1 mg/kg) plus nivolumab (3 mg/kg) regimen (10.9%) [15]. The median number of CKI doses administered across all CKI regimens was thirteen. The minimum number of doses was four and the maximum number of doses was sixty-four (Table 2). This outlier had an increased number of doses due to a previous oncologist opting to continue treatment indefinitely despite the patient achieving a complete remission. Upon transfer to our center, treatment was discontinued.

Ten of the forty-six individuals (21.7%) initially progressed during CKI treatment. These patients were converted to a CR with the addition of targeted therapies in addition to continuation of a PD-1 antibody treatment [16,19]. All progressing patients underwent NextGen molecular oncogene sequencing to identify potential targeted therapy options. As a result of this testing, four individuals were treated by the addition of dabrafenib (40.0%), two individuals were treated with dabrafenib plus trametinib (20.0%), two individuals were treated with trametinib monotherapy (20.0%), and one individual received encorafenib (10.0%). One individual was treated with sunitinib (10.0%). Another outlier patient had pseudoprogression with the persistence of subcutaneous lesions and the possible development of new sclerotic bone lesions. Treatment was continued for a total of 38 doses of CKI due to stable radiologic findings. Eventually, biopsies of persistent lesions established a pathologic complete response and treatment was discontinued.

### 3.3. Response Assessment

The progression-free survival (PFS) in patients who achieved a complete response on two sequential scans and then discontinued therapy was assessed from the end of therapy (EOT). PFS was 97.5% at 1 year and 94.7% at 30 months, with a median duration of follow-up of 26 ± 14 months from the EOT (Figure 1). Patients who did not achieve a CR had a markedly inferior PFS (median 5.5 months, *p* < 0.0001). It should be noted that the apparent drop-off in PFS in complete response patients after 30 months of follow-up from the end of treatment reflects a rapid decline in the number of individuals at risk due to the length of follow-up. Forty-two individuals have achieved a durable CR and have maintained an ongoing remission to date.

The median overall survival (OS) of 46 individuals who discontinued immunotherapy has not been reached. The OS continues to be 100% at a 4-year median follow-up (Figure 2). In contrast, the median OS was 8.8 months in patients not achieving a CR (*p* < 0.0001). Four individuals who achieved a complete response (8.7%) eventually relapsed. The median time to relapse was 27.4 ± 11.9 months (±SD). One of these four patients is currently undergoing additional adjuvant therapy after the resection of a lymph node recurrence. Another patient was salvaged with additional immunotherapy and achieved a second CR but subsequently developed a new high-risk primary melanoma. This patient is currently undergoing an additional adjuvant treatment. A third patient relapsed with metastatic disease and is being reinduced with an ipilimumab plus nivolumab treatment. The fourth patient progressed despite retreatment and passed away in hospice care. Two patients eventually died: one died from recurrent melanoma, and the other died from known severe coronary artery disease. A swim-lane plot of the time on therapy and their period of response to therapy is shown (Figure 3). The duration of therapy (orange) for all patients was shorter than the duration of their unmaintained remission after treatment discontinuation (blue).

### 3.4. Toxicity

Of the forty-six patients who achieved a CR, twenty-seven (58.7%) experienced significant treatment-related toxicities. Acute toxicities were defined as lasting less than six months, while chronic toxicities were defined as lasting more than six months after treatment discontinuation. Seventeen individuals experienced only acute toxicity (37.0%) and ten individuals experienced chronic toxicity (21.7%). The specific individual toxicities experienced by these patients are described in Table 2. Chronic toxicities included endocrinopathy (hyper- and hypothyroidism, and hypopituitarism), arthritis, fatigue/muscle weakness, pleural effusion, neuropathy, and mild creatinine elevation. Nineteen patients (41.3%) required additional medications and treatments to manage acute and chronic toxicity, including steroids, infliximab, antihistamines, endocrine hormone replacement, antipyretics, and in rare cases, other immunosuppressive agents. Most toxicities experienced were controlled in the outpatient setting. However, four individuals (8.7%) required hospitalization because of Grade III/IV toxicities. There were no treatment-related fatalities. None of patients required treatment discontinuation due to toxicity.

## 4. Discussion

Despite improvements in PFS and OS with advances in CKI therapy, there is no standard approach to treatment discontinuation in responding patients. It has become clear from multiple clinical trials that 20–40% of CKI-treated patients with metastatic melanoma achieve complete remissions and long-term survival [6,7]. Whether treatment can be safely discontinued in responding patients or should be continued has remained controversial, as it has been difficult to estimate the potential risk of relapse in such patients. Clinical trials have had significant variability in their design. Consequently, they did not provide adequate information about treatment discontinuation [20]. In addition, these clinical trials have highly selective entry criteria that may not represent real-world populations.

In the Checkmate 067 study, patients were allowed to continue therapy indefinitely. A decision to discontinue treatment was generally based upon disease progression, unacceptable toxicity, or a patient decision to withdraw. In the Keynote 001 study of pembrolizumab, a maximum treatment duration of two years was permitted [7]. Again, the decision to discontinue treatment was frequently based upon disease progression, intolerable toxicity, and a patient or investigator decision to withdraw.

These reports led us to evaluate treatment discontinuation outcomes in a large community practice melanoma cohort, as we have employed a similar treatment discontinuation strategy as that proposed by Robert et al. [20] over the last 6 years. Our report describes the outcome of treatment discontinuation employing a variety of CKI treatment regimens, including nivolumab monotherapy, pembrolizumab monotherapy, combinations of PD-1 antibody treatment with targeted agents, and combined ipilimumab and nivolumab (both standard and alternative dosing schedules), demonstrating the generalizability of this strategy.

Planned treatment discontinuation has previously been reported in two published studies. Amendments to the Keynote 001 study of pembrolizumab treatment in metastatic melanoma eventually allowed treatment discontinuation after patients achieved a radiologic complete remission (CR), received at least six months of CKI therapy, and received at least two additional doses after the determination of CR [21]. At the discretion of the investigator, many patients who were tolerating treatment well often continued therapy for up to 2 years. Robert et al. described the outcome of CKI discontinuation in patients from this trial. These investigators found a low rate of relapse in patients who discontinued treatment. Patients who achieved complete remission on two sequential scans 3 months apart had a 90.9% recurrence-free survival with a two-year median follow-up [21].

Our own data were based upon continual treatment until a documented CR confirmed by two scans at least 3 months apart. In our study, the overall survival was 100% at 4 years, and only four patients (8.7%) have relapsed. The median duration of treatment and number of CKI doses administered was 13 doses over a span of 9.6 months. The minimum treatment duration was 2.4 months with 4 doses of CKI. This outlier patient experienced severe treatment toxicity and had significant treatment delays. However, this patient had two scans demonstrating complete remission before the discontinuation of CKI treatment. The maximum duration of therapy was 36.2 months with 64 doses. Our data demonstrate a high likelihood that patients who achieve a radiographic or pathological complete response will remain in a durable remission. This approach was associated with a low late relapse rate (8.7%) and allowed a significant shortening of treatment duration. In several patients with protracted stable findings on radiographs, biopsies were needed to establish a complete response.

A recent publication by Betof Warner et al. evaluated the elective treatment discontinuation of PD-1 inhibitors in patients with metastatic melanoma [22]. Complete responses were seen in 25.8% of their 102 patients, who then electively discontinued treatment. These investigators reported that a surprisingly high percentage of patients relapsed (27.9%). In this study, most relapses occurred within the first 2 years. These investigators also found that only 15% of patients who relapsed responded to subsequent retreatment with anti-PD-1 agents [22].

Asher et al. reported 106 patients that discontinued CKI treatment. Of these patients, 80 had achieved a CR, and 26 only had partial responses or stable disease. The specific criteria for treatment discontinuation were not clearly described, but this series included 56.6% of patients who discontinued therapy due to toxicity. In this study, there was a relatively high relapse rate (22.7% in patients discontinuing therapy in CR and a 57.7% relapse rate in patients who achieved only a PR or SD) [23].

Pokorny et al. reported on 52 patients of 480 who received PD-1 inhibitors and discontinued therapy at a median of 11.1 months. Of these patients, 13 (25%) achieved complete responses, 28 (53.8%) achieved partial responses, and 11 (21.2%) at best achieved stable disease. After a median follow-up of 20.5 months (range 3–49.2) from treatment discontinuation, 39 (75%) patients remained without disease progression, while 13 (25%) had progression (median PFS 3.9 months; range 0.7–30.9). Only 2/13 CR patients relapsed versus 11/39 non-CR patients (not reaching statistical significance). In a multivariable analysis, younger age, history of brain metastasis, and higher lactate dehydrogenase at the time of anti-PD-1 discontinuation were associated with recurrence [24]. The potential differences between these studies and our own data (and that of Robert et al.) are likely to be due to the differences in the discontinuation strategy employed [22]. For instance, Betof Warner required only one scan to determine a complete response. In addition, 13% of their “complete responders” were determined by a clinical determination rather than more rigorous definition via radiographs or biopsy. Their reported median duration of treatment after patients achieved a CR was 0 months. In the other two studies, precise radiologic criteria for treatment discontinuation were not clearly identifiable [23,24]. The lower relapse rate we report is likely to be due to a more stringent determination of CR (i.e., by objective radiographic criteria and biopsy confirmation in the setting of residual stable radiologic abnormalities) and the continuation of treatment with CKIs beyond the initial determination of CR until confirmed by a second scan 3 months later.

A German report of 55 patients also found that the discontinuation of therapy in patients with a partial remission or stable disease was associated with a significant decrease in progression-free survival compared with patients who were in complete remission [25]. In this multivariate analysis, the duration of treatment, melanoma type, body mass index, programmed-death ligand 1 expression, and lactate dehydrogenase levels did not significantly influence the risk of relapse [25]. Similar effects of treatment discontinuation in CR versus non-CR patients were also observed by the other investigators above [22,23,24]. Our data, furthermore, suggest that patients who eventually achieve a complete remission have a marked progression-free and overall survival advantage over patients with partial responses, stable disease, or progressive disease. This dichotomous response in CR versus non-CR patients is potentially testable in a prospective clinical trial.

A shorter treatment duration for metastatic melanoma is advantageous for a variety of reasons. A shorter duration of CKI exposure has the potential to decrease the risk of treatment-related adverse events. Most acute toxicities (e.g., rash and colitis) occurred within the first few months of treatment [26], and several of our patients developed chronic side-effects. Some of these side effects, such as endocrinopathy, pleural effusion, and arthritis, proved to be troublesome during ongoing CKI therapy. By discontinuing therapy, these toxicities were easier to manage and control.

A decreased duration of therapy is also likely to decrease the economic burden of treatment. Potluri et al. compared the melanoma-specific costs in Great Britain and Germany following treatment with ipilimumab plus nivolumab treatment, nivolumab monotherapy, or ipilimumab monotherapy. Patient-level resource utilization data for the three treatment cohorts were obtained from the CheckMate 067 trial. All melanoma-specific resources, including drugs (index, concomitant, and subsequent melanoma medications), office visits, emergency room visits, hospitalizations, lab tests, procedures, and surgeries utilized over a 48-month evaluation period after start of index treatment were included. The total per-patient costs incurred by patients with advanced melanoma in both countries over the 48-month period following treatment initiation was upwards of USD 300,000 per patient for all regimens. There was a modest cost advantage for combination therapy due to decreased subsequent treatment and care costs. Drug costs accounted for >85% of the total melanoma treatment costs [27]. Similar conclusions were reached in a Canadian study, which also included estimates for pembrolizumab costs [28]. With TT, treatment-related costs are also high. It is estimated that the combination of dabrafenib plus trametinib adds approximately USD 19,000 in drug costs per month of treatment [29]. With the previous standards of treatment duration lasting 2 years or more, the economic burden to patients and payers was substantial. It appears likely that decreasing the treatment duration in appropriate patients could result in an appreciable cost savings.

Beyond minimizing the economic burden of treatment, patient quality of life will also likely improve with minimizing treatment duration. Frequent office visits, lab visits, and scans result in considerable time constraints and stress that negatively affect patient quality of life. Early treatment discontinuation allows patients to return to their usual daily activities and employment sooner. Hopefully our data will help promote further studies of CKI treatment duration to improve patient care and quality of life.

## 5. Conclusions

Elective discontinuation of checkpoint inhibitor therapy in melanoma is feasible and relatively safe in patients with metastatic melanoma who have eventually achieved a confirmed complete remission, as verified by sequential scans or a biopsy. We found that most confirmed complete remissions remained durable after elective treatment discontinuation. A potential limitation is that this series represents a relatively small real-life retrospective series that should be confirmed in prospective trials We hypothesize that this approach could decrease potential drug toxicities, reduce treatment-related financial burden, and improve patients’ quality of life.

## Figures and Tables

**Figure 1 biomedicines-10-01144-f001:**
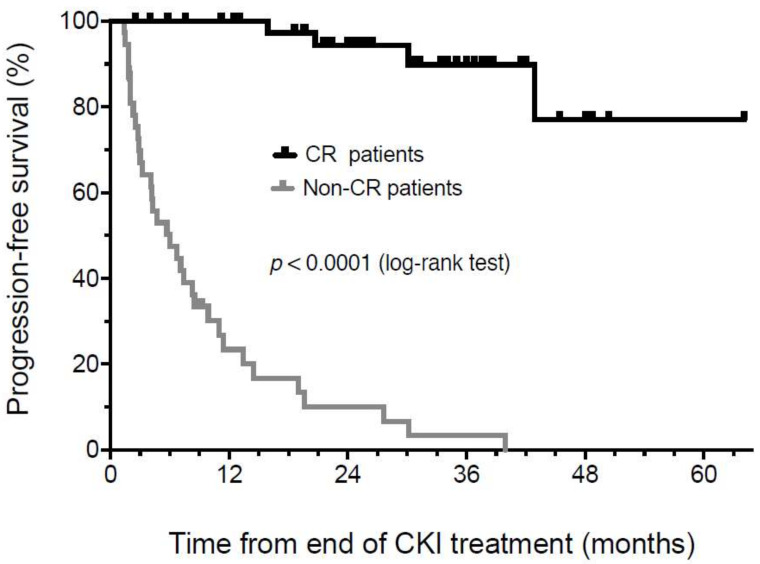
Kaplan-Meier analysis of progression-free survival from the time of treatment discontinuation in patients achieving a confirmed radiologic or pathologic CR compared to patients that never achieved complete remission.

**Figure 2 biomedicines-10-01144-f002:**
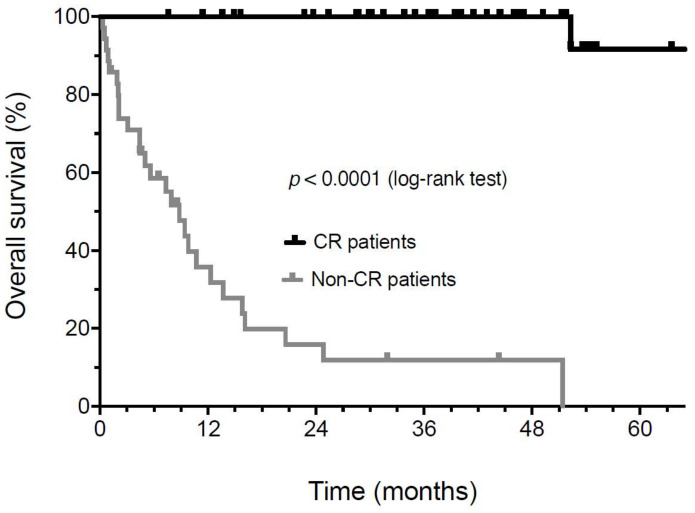
Kaplan-Meier analysis of overall survival from the time of treatment start in patients achieving a confirmed radiologic or pathologic CR compared to patients that never achieved complete remission.

**Figure 3 biomedicines-10-01144-f003:**
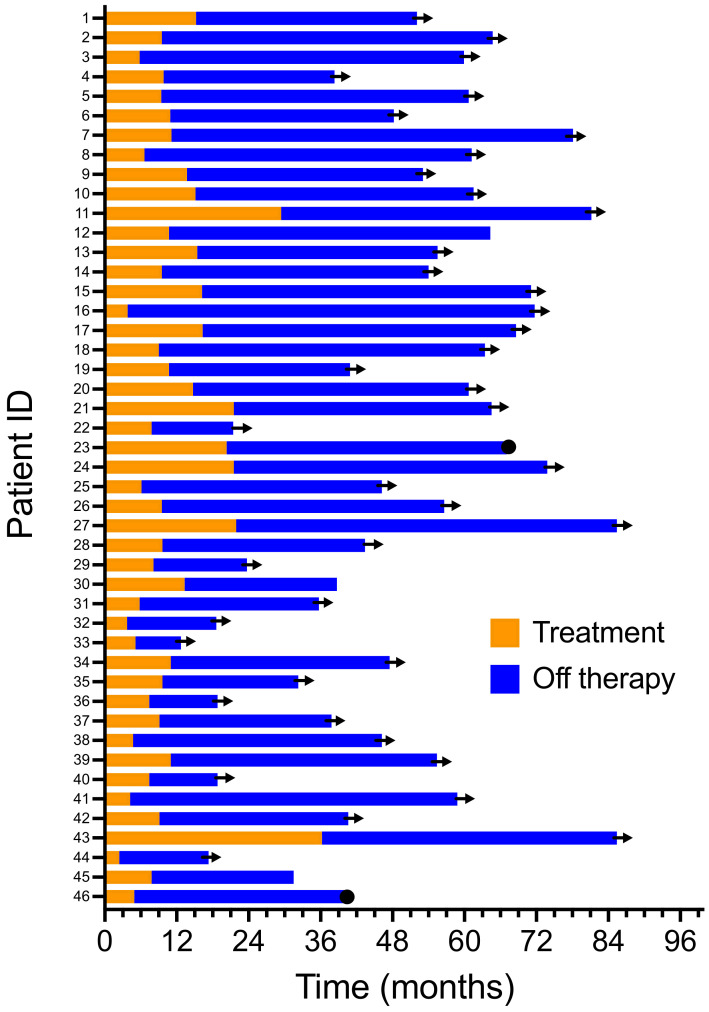
Swim-lane plot of duration of initial and second line therapy.

**Table 1 biomedicines-10-01144-t001:** Patient demographics.

UPN	Age	Sex	Race	Primary	Site of Metastases	Stage	KPS	LDH	Comorbidities
1	75	M	C	sinonasal	NA	T4, N0	100	N	-
2	73	M	C	trunk	brain, lung, adrenal	IVD	90	N	dementia
3	65	M	C	unknown	LN, adrenal	IVC	90	N	-
4	55	M	C	ear	in-transit, LN	IIIC	80	N	obesity, DM, HTN
5	67	F	C	arm	SQ	IVA	100	N	breast CA, RA, COPD
6	60	F	C	sinonasal	LN, abdomen, GI mucosa	IVC	90	N	IBS, hypoglycemia
7	47	M	C	leg	LN, SQ, lung, bone	IVC	90	N	-
8	48	F	C	leg	LN, SQ, lung, bone	IVA	100	N	NMSK, pseudothrombocytopenia
9	67	M	C	unknown	liver	IVC	100	N	prostate CA
10	71	F	C	arm	lung, brain	IVD	100	N	HTN, NMSK
11	63	M	C	trunk	sq, lung, bone	IVC	-	↑	DM, CAD, HTN
12	66	M	C	leg	in-transit, LN	IIIC	100	N	Parkinsonism, dementia
13	62	F	C	unknown	LN, lung	IVB	100	N	bronchitis, DJD
14	67	F	C	sinonasal	mediastinal LN, abdomen	IVC	80	N	COPD, DJD
15	70	F	C	subungual	extensive in-transit	IIIC	90	N	DJD, COPD, HTN
16	67	M	C	trunk	liver	IVC	80	N	Alzheimer’s, AK
17	64	M	C	trunk	skin, lung	IVB	90	↑	obesity, HTN
18	70	M	C	trunk	LN, SQ, lung, bone	IVA	100	N	HTN
19	26	M	C	trunk	LN, adrenal, bone	IVC	100	N	-
20	75	M	C	scalp	bowel, liver	IVC	100	N	DJD, gout, HTN, CAD
21	49	M	H	scalp	SQ, LN, adrenal, lung	IVA	90	N	obesity
22	79	M	C	trunk	bulky adenopathy	IIIB	90	N	obesity, hypothyroidism, BPH, sleep apnea, HTN, DM, AF, hypogonadism
23	84	M	C	face	liver, spleen	IVC	100	N	DM, HTN, Prostate CA, AF
24	77	M	H	ear	lung	IVB	-	N	DM, HTN, GIST
25	75	F	C	unknown	lung	IVB	90	↑	HTN, AVR, MVR, AF, CHF
26	68	F	C	trunk	bowel, lung	IVC	90	N	RA, HTN, osteoporosis
27	46	M	C	trunk	SQ, lung, bone	IVC	100	N	arthritis
28	57	M	C	trunk	chest wall	IVA	90	N	HTN, hypercholesterolemia
29	50	F	C	leg	LN, mesentery, adrenal	IVC	100	↓	-
30	59	M	C	trunk	lung, bone	IVC	90	N	arthritis, hypercholesterolemia, hypogonadism
31	68	M	C	trunk	sq, lung	IVB	100	N	AF, HTN
32	73	F	C	leg	SQ, LN	IVA	-	N	pulmonary fibrosis, psoriatic arthritis
33	56	M	C	trunk	SQ, pancreas, bone	IVC	90	N	DJD, CAD, HTN, DM
34	71	F	C	scalp	SQ, liver, lung, LN	IVC	90	N	Hypothyroid, DM
35	48	M	C	face	lung	IVB	100	N	hypercholesterolemia, HTN
36	77	M	C	unknown	pancreas	IVC	90	↑	hepatitis B, GERD, HTN, DJD
37	81	M	C	neck	lung	IVB	-	N	HTN, DM, CAD, MI, PVD
38	43	M	C	face	LN	IVA	100	N	PTSD, HTN, hypercholesterolemia, GERD
39	79	M	C	face	lung, SQ	IVB	90	N	DM, HTN, CKD, dialysis, NMSK
40	66	M	C	arm	SQ	IVA	100	↓	hypercholesterolemia, GERD
41	75	M	C	trunk	SQ, LN	IVA	90	N	DJD, NMSK, duodenal ulcer, gout
42	81	M	C	trunk	SQ, LN, lung	IVB	-	N	COPD, coccidiomycosis, fatigue
43	53	F	C	leg	LN, abdomen, brain	IVD	90	N	DJD, anxiety
44	64	F	C	scalp	SQ	IVA	100	↑	CLL, myositis
45	49	M	C	leg	extensive in-transit	IIIC	100	N	HTN, BPH, hypogonadism
46	66	M	C	leg	bone	IVC	100	N	CKD, HTN, DJD, NMSK

UPN, unique patient number; F, female; M, male; C, Caucasian; H, Hispanic; SQ, subcutaneous; LN, lymph nodes; N, LDH within normal range; ↑. LDH above upper limit of normal; ↓, LDH below lower limit of normal. DM, diabetes mellitus; HTN, hypertension; DJD, degenerative joint disease; CA, cancer; RA, rheumatoid arthritis; COPD, chronic obstructive pulmonary disease; IBS, inflammatory bowel disease; NMSK, non-melanoma skin cancer; CAD, coronary artery disease; MI, myocardial infarction; ARF, acute renal failure; PVD, peripheral vascular disease; AK, actinic keratosis; BPH, benign prostatic hyperplasia; AF, atrial fibrillation; GERD, gastroesophageal reflux disease; PTSD, post-traumatic stress disorder; MGUS, monoclonal gammopathy of undetermined significance; CKD, chronic kidney disease.

**Table 2 biomedicines-10-01144-t002:** Individual clinical outcomes of patients who discontinued therapy.

UPN	Regimen	CKI Doses	TT Added	TMB (/Mb)	PDL1 (%)	PFS (mo)	OS (mo)	CKI Toxicity	Current Status
1	N + I	17		3	-	21.7	36.9	dry mouth, hypothyroidism	NED
2	N	20	D	-	-	45.7	55.2	N	NED
3	N + I	12		-	-	48.2	54.1	elevated LFTs	NED
4	N + I	13		32	-	18.6	28.5	N	NED
5	N	17		-	21–30	42.0	51.3	N	NED
6	N + I	11		5	-	26.4	37.3	infusion rxn, melanoma-associated retinopathy	NED
7	N	22	D	-	-	55.9	67.0	N	NED
8	N	15		-	-	48.0	54.6	N	NED
9	N + I	16		-	0	41.1	39.4	colitis, RA flare, rash, dizziness	NED
10	P	20		-	5	31.3	46.4	N	NED
11	N + I	38	T	-	0	22.3	51.8	N	NED
12	P + RT	11	D + T	-	>1	42.9	53.6	arthralgias	PD
13	P	21		-	-	24.7	40.1	N	NED
14	P	10		-	-	47.1	44.5	facial swelling, dizziness, and dehydration	NED
15	P	20	T	-	<1	38.7	54.9	uveitis, vitiligo, hypopituitarism	NED
16	N	8		-	-	64.1	67.9	hypothyroidism, pruritus	NED
17	N + I + GM	24		-	-	36.0	52.3	hypopituitarism, hypophysitis	NED
18	N	19		-	-	45.4	54.4	C diff, vitiligo	NED
19	N + I	13	E	5	-	19.5	30.2	chills/sweats, diarrhea	NED
20	N	26	S	47	0	31.3	46.0	fatigue, muscle weakness	NED
21	N + I	24	D + T	25	-	21.5	43.0	N	NED
22	P	10		-	50–60	5.8	13.6	vertigo, lightheadedness	D-other
23	P	28		-	-	26.5	46.8	N	NED
24	P	24		-	-	30.8	52.3	N	NED
25	P + CK	9		155	-	34.1	40.1	low back pain, pleural effusion	NED
26	P	14		-	-	37.6	47.1	rash, joint pain	NED
27	N + I	27	D	-	>1	41.5	63.5	necrotizing granulomas, panhypopituitarism, eye pain	NED
28	N + I	9		32	-	24.2	33.8	colitis, hypothyroidism	NED
29	N + I	10		8	-	11.2	15.6	hypothyroidism, rash, pruritus, fatigue, myalgias, dizziness	NED
30	N + I	16		3	0	28.7	25.4	N	PD
31	N + I	7		50	-	24.0	29.9	rash, pruritic, panhypopituitarism	NED
32	N + I	6		24	2	11.2	14.9	rash, pruritis, psoriatic arthritis	NED
33	N + I	8		9	5	2.5	7.6	dry mouth	NED
34	N + I	13		-	-	25.4	36.5	N	NED
35	N + I	11		139	-	13.1	22.7	fever	NED
36	N + I	10		106	40	4.0	11.4	rash, pruritus, azotemia, dry mouth	NED
37	N + I	6		-	-	19.7	28.7	diarrhea, fatigue, muscle weakness, diplopia, dysphagia, dehydration, SOB, psoriasis flare	NED
38	N + I	8		-	-	36.8	41.5	N	NED
39	N	20		-	<1	33.4	44.4	N	NED
40	N	9		39	1	4.0	11.4	N	NED
41	P	8		-	<1	50.4	54.6	N	NED
42	P	13		52	-	22.4	31.5	necrotizing granulomas	NED
43	N	64	D	6	-	13	49.2	N	NED
44	N + I	3		9	-	12.5	14.9	hepatitis, rash and arthralgias, adenopathy splenomegaly	NED
45	N + I	10		57	1	15.9	23.7	N	PD
46	P	8		5	<1	30.1	35.0	N	DOD

UPN, unique patient ID; CKI, checkpoint inhibitor; TT, targeted therapy; OR, objective response; PFS, progression-free survival from end of immunotherapy; OS, overall survival (from start of therapy); N, nivolumab, I, ipilimumab; P, pembrolizumab; D, dabrafenib; T, trametinib, E, encorafenib; S, sunitinib; GM, GM CSF; RT, superficial electron beam radiotherapy to skin in-transit metastases; CK, cyberknife ablation of a solitary lung lesion; CR, complete response; PD, disease progression; PFS, progression-free survival; OS, overall survival; N, none, NED, no evidence of disease; D-other, died of other causes; DOD, died of disease.

## Data Availability

De-identified data supporting this manuscript are available upon reasonable request to the corresponding author.

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
