# Peer review of "Outcome of Elective Checkpoint Inhibitor Discontinuation in Patients with Metastatic Melanoma Who Achieved a Complete Remission: Real-World Data"

_biomedicines, 2022, doi:10.3390/biomedicines10051144_

Round 1

Reviewer 1 Report

In the present manuscript, Perez et al. present data from a retrospective chart review of patients with melanoma successfully treated with immune checkpoint inhibitor therapy who electively discontinued therapy (after 2 negative scans, 3 months apart). After a median duration follow-up of just over 2 years out of a cohort of 132 immune checkpoint inhibitor-treated melanoma patients, PFS data indicates 97.5 and 94.7 % of patients have ongoing response to treatment at 1 yr and 3 yr, respectively.

The paper is nicely written and clearly presented, with a nice summary of key and recent publications in the field directed at understanding the impact of elective immunotherapy discontinuation. The topic is of great interest to the field and would be of interest to the readership of this journal. However, the manuscript suffers from one major and several minor issues (detailed below). For this manuscript to be considered appropriate for publication, the following issues should be addressed.

Major concerns:

  1. Non-CR patient outcomes - The outcomes for patients with a CR (n=46) is excellent. How about non-CR patients (i.e., the other 86 patients)? Can the non-CR patients be subdivided into

Minor concerns/suggestions:

  1. Plotting the CR outcome data from the authors cohort on the same K-M curve side-by-side with the Robert et al. and Betof Warner et al. data would make for a compelling visualization
  2. Determining the correlation (or lack thereof) between beween duration of treatment and duration of response in CR versus non-CR patients would support the authors claim that more treatment does not necessarily yield better outcomes, and that the proposed treatment discontinuation strategy could yield similar or better outcomes with fewer
  3. Can the authors comment on any differences in outcomes for CR patients treated with ipi+nivo vs nivo/pembro single agent? It would appear a CR regardless of treatment strategy is associated with excellent outcomes.
  4. Can the authors comment on the impact of immune suppression before, during, or after treatment discontinuation?
  5. Can the authors comment on the outcomes for the subset of CR patients who developed acute toxicity and were not able to be retreated?

Author Response

Reviewer 1

In the present manuscript, Perez et al. present data from a retrospective chart review of patients with melanoma successfully treated with immune checkpoint inhibitor therapy who electively discontinued therapy (after 2 negative scans, 3 months apart). After a median duration follow-up of just over 2 years out of a cohort of 132 immune checkpoint inhibitor-treated melanoma patients, PFS data indicates 97.5 and 94.7 % of patients have ongoing response to treatment at 1 yr. and 3 yr., respectively.

The paper is nicely written and clearly presented, with a nice summary of key and recent publications in the field directed at understanding the impact of elective immunotherapy discontinuation. The topic is of great interest to the field and would be of interest to the readership of this journal. However, the manuscript suffers from one major and several minor issues (detailed below). For this manuscript to be considered appropriate for publication, the following issues should be addressed.

Major concerns:

  1. Non-CR patient outcomes - The outcomes for patients with a CR (n=46) is excellent. How about non-CR patients (i.e., the other 86 patients)? Can the non-CR patients be subdivided into

We appreciate the reviewer’s excellent suggestion.  We have provided comparison data on non-CR patients (PR, SD, and PD patients) in our Kaplan Meier plots of PFS and OS (Figures 1 and 2).  It should be noted that unlike many studies that evaluated objective responses after only 3 months of therapy, our data represents maximal responses (assessed at > 9 months after the start of treatment unless patients progressed and died prior to 9 months)(now better described p3, also in materials and methods p4).  It is interesting to note that the overall response pattern appeared to be dichotomous:  Patients who achieved a best response of CR at 9-12 months had a highly significant improvement in PFS and OS compared to non-CR patients (described p4, 5).  This is brought out in the discussion (p 8), as we would propose the testable hypothesis that in some of the benchmark studies (e.g., Keynote 001 and Checkmate 067) patients who had PR or stable disease at 3 months with long term PFS and OS may have been patients who were converted to CR with ongoing therapy.  In our data, we could not separately evaluate PR or stable disease patients since they represented only a very small fraction of patients (2-4% of overall patients), as they either appeared to eventually convert to CR or developed progressive disease.

Minor concerns/suggestions:

  1. Plotting the CR outcome data from the authors cohort on the same K-M curve side-by-side with the Robert et al. and Betof Warner et al. data would make for a compelling visualization

While this sort of back-of-the-napkin data comparison is sometimes performed to informally compare outcomes of clinical trials, it is statistically questionable.  There are likely to be very significant differences between patient populations and therefore outcomes in comparing rigorous patient selection in a clinical trial compared to real world data (unselected patients, but with potential referral bias).  The likely variation in these disparate patient populations limits the validity of any implied comparison.  We would instead suggest that this sort of comparison be approached in a prospective randomized trial.

  1. Determining the correlation (or lack thereof) between duration of treatment and duration of response in CR versus non-CR patients would support the authors claim that more treatment does not necessarily yield better outcomes, and that the proposed treatment discontinuation strategy could yield similar or better outcomes with fewer

It should be noted that all non-CR patients in our study continued ongoing therapy until hospice enrollment or death.  These patients did not ever discontinue therapy for any significant period, despite inferior PFS and OS.  Thus, their inferior outcome was not due to less treatment compared to CR patients who were treated for a number of immunotherapy doses but were able to discontinue treatment for prolonged periods.

  1. Can the authors comment on any differences in outcomes for CR patients treated with ipi+nivo vs nivo/pembro single agent? It would appear a CR regardless of treatment strategy is associated with excellent outcomes.

The frequency of complete responses is modestly higher with combined ipilimumab/nivolumab compared to PD-1 monotherapy (especially in BRAF mutant melanoma) as has been published in Checkmate 067.  Approximately 22% of our patients entered a complete remission after addition of targeted agents to PD-1 monotherapy after failure of 1st line ipilimumab/nivolumab (often hyperprogression at initial scan) or following failure of single agent PD-1 therapy.  The key to successful treatment discontinuation in any of these groups (ipi+nivo, single agent PD1 or PD1 plus targeted therapy) appeared to be a confirmed complete response on two sequential scans

  1. Can the authors comment on the impact of immune suppression before, during, or after treatment discontinuation?

Approximately 20-30% of our patients developed grade 3-4 toxicities (such as rashes, diarrhea, occasionally other Immune Related Adverse events).  These were always managed with aggressive use of glucocorticosteroids and in some cases addition of secondary immunosuppressive agents (e.g., infliximab for colitis).  This was almost always managed the outpatient setting.  Hospitalizations were rare (only 4 patients). Patients who developed toxicity had at least as high a chance of achieving a complete remission as non-steroid or non-infliximab treated patients despite immunosuppressive therapy (see Table 2 for patients with and without toxicity).  Twenty of 41 patients achieved a CR without toxicity while 22 of 41 had toxicity requiring at minimum steroid administration (not statistically different).

  1. Can the authors comment on the outcomes for the subset of CR patients who developed acute toxicity and were not able to be retreated?

Since checkpoint inhibitor toxicity is aggressively monitored and treated in our outpatient clinic (while only grade 1 or 2), complete and early resolution is frequent.  In most patients, treatment almost always can be resumed.  None of the CR patients required treatment discontinuation for toxicity.  Of the non-CR patients represented in this study, only 1 or 2 developed a type of toxicity that prevented further therapy (e.g., neurologic syndromes like CIDP, blistering skin diseases).  There were no cases of myocarditis or pneumonitis.  None of the non-CR patients developed durable responses.  We are aware that durable remissions have occasionally been observed after severe toxicity and treatment discontinuation, even after a single cycle of ipilimumab/nivolumab.

Reviewer 2 Report

In this manuscript Leanne Perez and colleagues reported a retrospective clinical experience of metastatic melanoma patients undergoing elective treatment discontinuation of checkpoint inhibitors (CKIs). Understanding how to manage elective CKI discontinuation it is still a matter of debate. Importantly, proper therapy discontinuation could ensure long-term outcomes with the advantage of reducing the immunotherapy-related toxicities and financial burdens associated with prolonged CKI treatment.

As for other published studies this is only a small-scale real-life retrospective observation which should be confirmed in prospective trials. This should be emphasized as a limitation in their conclusion.

The Authors defined as elective discontinuation strategy the occurrence of Complete Response (CR) documented by 2 negative scans 3 months apart. On their opinion this represent the major difference compared to other studies in which the CR was determined only by less stringent clinical evaluation.

However, many recent reports are missing and have not been discussed, particularly: Nethanel Asher et al 2021 and Rebecca Pokorny et al 2021. The Authors should comment these works.

The Authors should better discuss the implication of previous therapy exposure in their cohort and specify how many patients were treatment-naïve.

Minor points:

Table 1: LDH should be expressed as value

Table 2: Please specify RT and the difference between P/RT and P+RT

Author Response

Reviewer 2

In this manuscript Leanne Perez and colleagues reported a retrospective clinical experience of metastatic melanoma patients undergoing elective treatment discontinuation of checkpoint inhibitors (CKIs). Understanding how to manage elective CKI discontinuation it is still a matter of debate. Importantly, proper therapy discontinuation could ensure long-term outcomes with the advantage of reducing the immunotherapy-related toxicities and financial burdens associated with prolonged CKI treatment.

As for other published studies this is only a small-scale real- life retrospective observation which should be confirmed in prospective trials. This should be emphasized as a limitation in their conclusion.

This caveat has been added to conclusion section (p 15)

The Authors defined as elective discontinuation strategy the occurrence of Complete Response (CR) documented by 2 negative scans 3 months apart. On their opinion this represent the major difference compared to other studies in which the CR was determined only by less stringent clinical evaluation.

Another important difference is that the definition of partial response, stable disease or progressive disease was assessment of the best response at > 9 months follow-up (better described on page 4-5)

However, many recent reports are missing and have not been discussed, particularly: Nethanel Asher et al 2021 and Rebecca Pokorny et al 2021. The Authors should comment these works.

These reports have been cited and added to the discussion (p8). 

The Authors should better discuss the implication of previous therapy exposure in their cohort and specify how many patients were treatment naïve.

In our institution all metastatic melanoma patients have been treated with immunotherapy first for many years due to delays in obtaining Next-Gen molecular testing of their tumor samples.  Thus, all patients were treatment naïve at the start of therapy (Added to p4).  It should be noted that if patients progressed following initial checkpoint-inhibitor therapy, 2nd line therapy (e.g., addition of targeted agents to continued PD-1 monotherapy in patients with BRAF, NRAS and NF1 mutations converted approximately 22% of previously progressing patients to CR as we have previously published)[1,2]

Minor points:
Table 1: LDH should be expressed as value

We chose not to provide numerical values, as there are two different significantly different normal ranges employed by the lab facilities that usually process samples for our patients.  This would likely to be confusing to readers if numerical values we provided

Table 2: Please specify RT and the difference between P/RT and P+RT

Records of these patients were individually reviewed.  Clarified in legend Table 2.  One patient was inadvertently included as having radiation therapy when this was used to treat a metachronous localized prostate cancer, not the metastatic melanoma.  This has been corrected in the table.

  1. Samlowski, W.; Adajar, C. Cautious addition of targeted therapy to PD-1 inhibitors after initial progression of BRAF mutant metastatic melanoma on checkpoint inhibitor therapy. BMC Cancer 2021, 21, 1187-1199, doi:10.1186/s12885-021-08906-1.
  2. Hilts, A.; Samlowski, W. Cautious Addition of MEK Inhibitors to PD-1 Antibody Treatment in Patients with NRAS or NF1 Mutant Metastatic Melanoma Failing Initial Immunotherapy. Annals of Case Reports 2022, 7, 795-805, doi:www.doi.org/10.29011/2574-7754.100795.

Round 2

Reviewer 1 Report

In their revised manuscript, Perez et al. have reasonably and appropriately addressed major and minor concerns from both reviewers. Highlighting the limitations of the study, including additional references, and modifying figures 1 and 2 have improved the manuscript. 

Reviewer 2 Report

The Authors replied to all the comments and adequately revised the manuscript.

The revised manuscript is acceptable in its present form.